# Radiomics: The New Promise for Differentiating Progression, Recurrence, Pseudoprogression, and Radionecrosis in Glioma and Glioblastoma Multiforme

**DOI:** 10.3390/cancers15184429

**Published:** 2023-09-05

**Authors:** Mohammadreza Alizadeh, Nima Broomand Lomer, Mobin Azami, Mohammad Khalafi, Parnian Shobeiri, Melika Arab Bafrani, Houman Sotoudeh

**Affiliations:** 1Physiology Research Center, Iran University of Medical Sciences, Tehran 14496-14535, Iran; dr.alizadeh.d@gmail.com; 2Faculty of Medicine, Guilan University of Medical Sciences, Rasht 41937-13111, Iran; nima.broomand@gmail.com; 3Student Research Committee, Kurdistan University of Medical Sciences, Sanandaj 66186-34683, Iran; mobinaz98@yahoo.com; 4Radiology Department, Tabriz University of Medical Sciences, Tabriz 51656-65931, Iran; mohammadkhalafi4287@gmail.com; 5School of Medicine, Tehran University of Medical Sciences, Tehran 14167-53955, Iran; parnian.shobeiri@gmail.com (P.S.); melika.arab92@gmail.com (M.A.B.); 6Department of Radiology and Neurology, Heersink School of Medicine, University of Alabama at Birmingham (UAB), Birmingham, AL 35294, USA

**Keywords:** glioma, glioblastoma multiform (GBM), radiomics, MRI, PET, tumor progression, tumor recurrence, pseudoprogression, radionecrosis

## Abstract

**Simple Summary:**

Progression/recurrence, pseudoprogression, and radionecrosis are all scenarios that can be expected during the treatment course of glioma and GBM. Although MRI, PET, CT, and MRS have shown some capabilities in differentiating these conditions, there is still a considerable need for the emergence of state-of-the-art techniques to assist field professionals. Here, we introduce radiomics, a process that extracts many features from medical images using data characterization algorithms and a promising tool to differentiate these scenarios. The results could significantly impact patients’ care by enhancing the understanding and accuracy of post-treatment follow-ups in brain cancer patients.

**Abstract:**

Glioma and glioblastoma multiform (GBM) remain among the most debilitating and life-threatening brain tumors. Despite advances in diagnosing approaches, patient follow-up after treatment (surgery and chemoradiation) is still challenging for differentiation between tumor progression/recurrence, pseudoprogression, and radionecrosis. Radiomics emerges as a promising tool in initial diagnosis, grading, and survival prediction in patients with glioma and can help differentiate these post-treatment scenarios. Preliminary published studies are promising about the role of radiomics in post-treatment glioma/GBM. However, this field faces significant challenges, including a lack of evidence-based solid data, scattering publication, heterogeneity of studies, and small sample sizes. The present review explores radiomics’s capabilities in following patients with glioma/GBM status post-treatment and to differentiate tumor progression, recurrence, pseudoprogression, and radionecrosis.

## 1. Introduction

Glioma is the name for all cancers that are thought to come from glial cells. They are also the most common type of brain and spinal cord cancer. About 57% of all gliomas are glioblastoma (GBM), the most common and life-threatening brain tumor. Additionally, 48% of all primary aggressive central nervous system (CNS) tumors are also GBM [1,2,3]. In 2021, the World Health Organization (WHO) classified glioma and glioblastoma based on key molecular biomarkers; this describes neoplastic entities and makes it much less critical for tumor classification to depend on morphologic features. Under the 2021 update, adult diffuse gliomas are divided into three main disease groups based on molecular markers: IDH-mutant, 1p/19q codeleted oligodendroglioma; IDH-mutant, non-codeleted astrocytoma; and IDH-wildtype glioblastoma [4,5].

Surgery with maximum safe reaction is the standard treatment utilized to treat GBM. The aim of resection is a gross total resection without risking the patient’s functional status. However, complete resection is often not practical, given the infiltrative behavior of gliomas. So, surgery is usually followed by chemotherapy and/or radiation treatment. Today, most people with glioblastoma receive a mix of treatments before and after surgery [6,7]. These include radiotherapy (RT) alone or with chemotherapy. After surgery, standard treatment gives 60 Gy in 2 Gy doses over six weeks, along with temozolomide (TMZ). Currently, there are three Food and Drug Administration (FDA)-approved medications to treat GBM: TMZ, bevacizumab, and BCNU (carmustine). TMZ is currently the most common FDA-approved chemotherapy drug for treating glioblastoma [8]. An increase in median survival rate has been observed with these combination therapeutics. Other treatment protocols are now mainly in the research phase, including tumor-treating fields (TTFields), vaccine-based immunotherapies, and oncolytic viral T-cell immunotherapy [8].

IDH-mutant astrocytoma and 1p/19q codeleted oligodendroglioma are usually treated by maximum surgical resection. The subsequent treatment depends on the grade of the neoplasm on histopathology. IDH-mutant astrocytoma grade 2 and 1p/19q codeleted oligodendroglioma grade 2 without residual tumor on fluid-attenuated inversion recovery (FLAIR) magnetic resonance imaging (MRI) will be followed by surveillance MRI. IDH-mutant astrocytoma grade 2 with a residual tumor on FLAIR MRI, 1p/19q codeleted oligodendroglioma grade 2 with residue on FLAIR MRI, grades 3 and 4 IDH-mutant astrocytoma, and grade 3 1p/19q codeleted oligodendroglioma will be treated by chemoradiation [9].

Radionecrosis causes additional lesions resembling tumor progression or recurrence during imaging follow-up. Correctly differentiating these lesions is crucial since their therapeutic paradigms differ [10,11].

MRI and positron emission tomography (PET) approaches have shown the capability of differentiating these scenarios [12,13,14]. There is still no consensus on their effectiveness due to the lack of investigations and heterogeneity in scanning protocols. In addition, these approaches have faced severe restrictions in correctly differentiating these approaches [15,16].

All types of glioma status post-treatment need to be followed by MRI for diagnosis of recurrence and or progression. The interpretation of MRI findings in glioma after treatment is challenging, given the fact that there is an overlap between post-treatment signal alteration and recurrence/progression.

The post-treatment glioma MRI can show the following conditions:A.Pseudoprogression: A transient enlargement of tumoral abnormal signal intensity and enhancement after chemoradiation (usually within six months after treatment) caused by inflammation, edema, damage to the endothelium, blood–brain barrier (BBB) disruption, and oligodendroglial injury after treatment. It is more common within 3 months after completion of therapy, but it can occur years after treatment. Moreover, it is more common in O6-methylguanine-DNA methyltransferase (MGMT)-methylated tumors, particularly when treated with TMZ [15]. Patients are usually stable clinically. Pseudoprogression is generally associated with a longer survival rate [17,18,19]. Pseudoprogression (psp) arises from a pronounced local tissue reaction involving inflammation, edema, and abnormal vessel permeability, leading to new or increased contrast-enhancing lesions. While less severe cases may subside without additional treatment, more severe cases can progress to true treatment-related necrosis over time [20]. The differentiation between tumor progression and pseudoprogression presents a significant challenge, and advanced imaging techniques such as advanced MRI and PET imaging show promise in improving the accuracy of this differentiation [21]. Follow-up MRI scans, conducted 4 to 8 weeks after the initial scan, are commonly utilized to aid in distinguishing between the two conditions [20].B.Radiation necrosis: Radiation can cause radiation-induced neurotoxicity in the brain parenchyma, secretion of tumor necrosis factor-alpha (TNF-α), endothelial damage, damage to the BBB, glial and subsequence worsening of edema, and the enhancement/evolution of new areas of abnormal enhancement mimicking recurrence/true progression. Radiation necrosis (RN) usually happens 3–12 months after RT in approximately 3–24% of adult brain tumors but can be seen up to several years and even decades after RT [15]. Histological examination reveals necrosis, edema, gliosis, endothelial thickening, hyalinization, fibrinoid deposition, thrombosis, and vessel occlusion. These pathological criteria distinguish RN from other glioma-related conditions [22]. TNF-α is the primary cytokine released following radiation. Other cytokines that cause endothelial cell death, astrocyte activation, and BBB permeability are upregulated by TNF-alpha [23,24]. The imaging features of radiation-induced necrosis present challenges in differentiation, as the contrast-enhancing mass on T1-weighted imaging with gadolinium appears similar to tumor progression using conventional MRI techniques [25,26].C.Recurrence: Recurrence is one of the leading causes of death in glioma and GBM [27]. Recurrence timelines can demonstrate substantial variability. A study centered on patients diagnosed with low-grade glioma highlighted that a proportion of 28% had recurrence events within two years subsequent to their main surgery. Conversely, a more substantial proportion, 72%, witnessed recurrence events after this two-year threshold [28]. The timing of recurrence is influenced by the grade of the glioma. High-grade gliomas like glioblastoma have a high recurrence rate, with most recurrences found near the original tumor site [29,30].

The gold standard for diagnosing recurrent glioma is histologic confirmation. Still, the decision to perform a biopsy must weigh the diagnostic value against procedural risks, especially considering potential complications from previous treatments like radiation or chemotherapy. During the first six months post-treatment, radiographic changes might indicate pseudoprogression, leading many doctors to opt for regular MR imaging instead of immediate biopsy. However, suppose a new lesion emerges on MR images, particularly outside the initial high-dose radiotherapy zone or 6–12 months postradiotherapy. In that case, it might favor tumor recurrence and prompt further actions even without histologic proof of recurrence [30]. Noteworthily, compared to primary tumors, recurrent gliomas more frequently exhibit features such as copy number variations (CNV), combined IDH1 and TERT mutations, compromised cell cycle signaling pathways, and low tumor mutational burden (TMB) [31]. Remarkably, upon recurrence, gliomas might display variations in their histological characteristics. A previously identified low-grade glioma might escalate to a high-grade form. This evolution could be linked to prior therapeutic interventions, such as the intensity of radiation or chemotherapy sessions [32].


D.True progression: Malignant progression alongside the recurrence of low-grade glioma primarily contributes to its treatment complications and poor prognosis [33]. Pathologically, true progression (TP) is characterized by neovascularization, the proliferation of tumoral cells, and the disruption of the BBB [20]. Numerous determinants, including genetic evolution, microenvironmental interplay, and histological features alterations, mark the progression of gliomas. Additionally, the presence or absence of IDH mutations plays a role in shaping the course of glioma progression, having implications for patient prognosis and the degree of cell malignancy [34]. The glioma’s molecular details and brain location play a critical role in determining its progression rate, which can span from a mere two years to well over ten years [35]. Of note, glioblastoma can exhibit different progression patterns, such as local, diffuse, distant, and multifocal [36]. Although several molecular markers have been identified to predict the progression of the glioma, a lack of standardized methods and insufficient clinical trials have hindered the practicality of this approach in clinical settings [37].


## 2. Current Differentiating Approaches in Glioma after Treatment

Imaging modalities greatly influence the primary diagnosis and post-therapeutic follow-up of brain gliomas and glioblastomas. Different modalities, such as MRI, computed tomography (CT), and PET, have been utilized in these scenarios [12,13,14] (Table 1). MRI is a commonly used modality in diagnosing and managing patients with glioblastoma or glioma and is the standard of care for the radiographic characterization of glioblastoma after treatment. New foci of enhancement within and around the resection cavity/radiation field can be seen in both recurrence and radiation necrosis. Usually, nodular boundaries with noticeable edges are more common in tumor recurrence, and blurred plumed borders favor RN. The corpus callosum’s involvement with midline crossing, subependymal dissemination, and numerous enhancing lesions promoted tumor recurrence over RN [26,38,39]. It is also stated that the subependymal enhancement could be the only predictive sign in cases of early progression [40]. If a new focus of enhancement appears after treatment but does not change or gets smaller over time, MRI helps show pseudoprogression [41]. Further, pseudoprogression appears as a self-resorbing focal contrast enhancement in this modality [42]. However, contrast enhancement can be both indicator of the therapy response or tumor relapse. It also manifests the increased permeability of the BBB, which is not specific to these scenarios [43]. The other issue is that GBM and anaplastic glioma may show no or minimal enhancement on MRI, limiting decisions determining invasion and the boundaries of the tumor [44].

In daily practice, the most common modality for differentiation between recurrence/progression and RN is dynamic susceptibility contrast (DSC) MR perfusion. It is well known that the relative cerebral blood volume (rCBV) is higher in tumor progression/recurrence than RN. However, radiation necrosis usually shows heterogeneity on diffusion-weighted imaging (DWI) images [45]. With apparent diffusion coefficient (ADC) values that are larger in necrotic tissue than in recurrent tumor tissue, DWI can distinguish recurrence from pseudoprogression. In contrast to pseudoprogression, tumor progression exhibits higher DSC-derived parameters such as peak height and the percentage of signal recovery [46]. However, neither DWI nor diffusion tensor imaging (DTI) offer enough details to distinguish pseudoprogression from TP reliably. Both DWI and ADC maps produce diverse signal intensities, with regions of reduced diffusion that might signify either highly cellular tumor areas or inflammatory processes [43,47,48,49]. Moreover, diffusion imaging can have limitations in resolving lesions with a mixture of recurrence and treatment necrosis since ADC values can be similar in both cases [50]. On top of that, there is a lack of validated diagnostic criteria on an individual level. This means there is no standard way to interpret DWI images, leading to differences in diagnosis between radiologists [51]. Variability in sequences from one center to another, between scanners in the same center, or even in a single scanner can lead to differences in image quality, affecting the accuracy of the diagnosis [52]. Equally important, Rcbv is only semiquantitative (hence the term relative CBV), and that model’s assumptions are violated when there is leakage of contrast agent from the intra- to the extravascular compartment, which is invariably the case in enhancing tumors [20].

MRS is another noninvasive diagnostic tool that measures biochemical changes in the brain, especially the presence of tumors. The choline/creatine ratio used in this method is a good predictor of differentiation between true progression and treatment-induced changes [53]. Recurrent brain neoplasms exhibit an elevation in choline (Cho) as a reflection of increased cell membrane turnover [54]. Moreover, features of radiation necrosis include a variable decrease in n-acetyl-aspartate (NAA), a lack of pronounced Cho elevation, and the presence of lipid-lactate peaks [55]. However, relatively large voxel sizes may lead to partial volume effects between recurrent tumors and treatment-induced changes, limiting its differential power [56]. Beyond that, due to low metabolite concentrations, many acquisitions are required, resulting in long scan times [22]. In the same vein, the need to exclude signal contamination from tissues adjacent to the tumor, such as lipids (from the scalp) and water (from ventricles), makes magnetic resonance spectroscopy (MRS) a challenging technique. Surgical clips also disrupt the local field homogeneity, affecting data quality [53]. Furthermore, there is a considerable overlap of metabolite peaks. This overlap can make distinguishing between different metabolites difficult and lead to misinterpretation of the results [57].

Different modalities of PET scans can also assist in differentiating these stages. Tumor recurrence will usually appear as metabolically active lesions on FDG-PET [48]. Also, radiation necrosis manifests as a metabolically inactive lesion on FDG-PET [48]. However, a lack of specificity and high background uptake in healthy brain tissue curb its ability in differentiating glioma progression from pseudoprogression [58]. On the other hand, amino acid PET is well suited for monitoring treatment response and diagnosing pseudoprogression because amino acid tracers can cross the blood–brain barrier [50]. It can specifically differentiate the cellular component of a tumor mass from inflammatory and necrotic lesions, providing an early response to therapy [59]. This modality can also differentiate between true progression and pseudoprogression, with higher Tbrmax and mean tumor-to-brain ratio (tbrmean) favoring true progression [60], while pseudoprogression manifests with a relatively lower uptake of the radioactive tracer [61]. Nevertheless, amino acid PET requires longer acquisition times than other PET imaging modalities [62]. Additionally, both progression and pseudoprogression can increase metabolic activity on PET scans, making it difficult to differentiate between them [63]. Another issue is that high uptake soon after radiotherapy may be treatment-related, which can be mistaken for recurrence [64]. Moreover, PET is a costly approach and lacks approval and reimbursement by national insurance, which limits its practicality in all settings [65]. Lastly, amino acid PET requires an on-site cyclotron due to its short half-life of 11C, further limiting its widespread usage. [15].

Combining these modalities provides a more accurate diagnosis in these scenarios [66]. In contrast to PET/CT, PET/MRI does not burden ionizing radiation. This can be specifically important when planning for multiple scanning sessions [66]. However, PET/MRI scanners are expensive and not widely accessible. Moreover, the PET/MR imaging parameter cutoff values are not standardized [16]. What is more, the combined approach needs a high scanning time, which may be divided into several daunting sessions [67].

Additionally, a review study established that conventional MRI and 18F-FDG PET possessed higher sensitivity, while thallium single-photon emission CT (SPECT) maintained superior specificity in distinguishing between progressive disease, pseudoprogression, and RN [68]. However, low photon flux, low anatomic resolution, and tracer uptake in the choroid plexus or pituitary limits the SPECTs data for diagnosing [10].

The current performance metrics of different imaging modalities in post-treatment glioma are summarized in Table 1.

**Table 1 cancers-15-04429-t001:** Current application and performance of different imaging modalities in post-treatment glioma.

Imaging Modality	Dx of Recurrence, Treatment Response (Radionecrosis), True Progression	DDX of Recurrence vs. Treatment Changes	DDX of True Progression vs. Radionecrosis	DDX of True Progression vs. Pseudoprogression
FDG PET	Recurrence: sensitivity = 78%, specificity = 88% [69].		Sensitivity = 0.86, specificity = 0.80, accuracy = 0.83, cut-off: maximum standardized uptake value (SUV max) = 1.9 [70]. ** Sensitivity = 79%, specificity = 70% [71].	
Amino acid PET		Sensitivity = 93%, specificity = 100%, accuracy = 93%, TBR = 1.6 [72]/ sensitivity = 0.88, specificity = 0.78, diagnostic odds ratio = 26, the area under the curve (AUC) of hierarchical summary receiver operating characteristic (HSROC) = 0.86 [73].		Sensitivity = 100%, specificity = 91%, accuracy = 96%, maximum tumor-to-brain ratio (tbrmax) cutoff = 2.3 [74]. Sensitivity = 100%, specificity = 79%, accuracy = 83% [75]/sensitivity = 84%, specificity, = 86%, accuracy = 85% [60].
Conventional MRI (T1, T2, FLAIR, T1 + C)	Recurrence: sensitivity = 0.36, specificity = 0.93, AUC = 0.75 [76]/sensitivity = 31.7%, specificity = 80%, PPV = 96.3%, NPV = 6.7% [77]. Early progression: sensitivity = 0.81, specificity = 0.69, AUROC= 0.79 [78].	Sensitivity = 8%, specificity = 91%, PPV = 25%, NPV = 73% [79].	Sensitivity = 88.9%, specificity = 33.4% [80]. Sensitivity = 38.1%, specificity = 93.3%, NPV = 41.8% [40].	
DWI/ADC	Treatment response: sensitivity = 0.71, specificity = 0.87 [53].	Sensitivity = 52.6–94.7%, specificity = 50–90% [81].		TR vs. pseudoprogression: sensitivity = 0.88, specificity= 0.85 [82]/ TR vs. pseudoprogression: sensitivity = 0.90, specificity = 0.82, accuracy = 0.93 [83].
MR perfusion	Treatment response: sensitivity = 0.87, specificity = 0.86 [53].	Sensitivity = 0.9, specificity = 0.88 [84]/sensitivity = 0.83, specificity = 0.85, AUROC = 0.91 [85]/sensitivity = 0.88, specificity = 0.88 [84].		Sensitivity = 0.88, specificity = 0.77, AUROC = 0.88 [86]/sensitivity = 0.85, specificity = 0.79, accuracy = 0.9 [82].
MR spectroscopy (Cho/NAA or Cho/Cr)	True progression: sensitivity = 71.2%, specificity = 90.2%, AUC = 0.792 [87]. Treatment response: sensitivity = 91%, specificity= 95% [53].	Sensitivity = 75.0%, specificity = 81.0%, accuracy = 79% [88].	Sensitivity = 60%, specificity = 45%, PPV = 16% NPV = 87% [89].	
Multimodal MRI (conventional sequences + DWI/ADC + MRP + MRS)		Sensitivity = 80.6%, specificity = 66.6% [90].		
PET/MRI	Recurrence: sensitivity = 97.14%, specificity = 93.33%, accuracy: 96% [91].		Sensitivity = 0.88, specificity = 0.79, AUC = 0.91 [73].	

** In order to maintain a more comprehensive and updated approach, some sections contain findings from more than one study. Abbreviations. Dx: diagnosis, DDx: differential diagnosis, SUV: standardized uptake value, RT: recurrence of tumor, FDG: fluorodeoxyglucose, PET: positron emission tomography, TBR: tumor-to-background ratio, AUC: area under the curve, HSROC: hierarchical summary receiver operating characteristic, FLAIR: fluid-attenuated inversion recovery, PPV: positive predictive value, NPV: negative predictive value, ADC: apparent diffusion coefficient, DWI: diffusion-weighted imaging, MRS: magnetic resonance spectroscopy.

Radiomics is a quantitative approach aiming to extract mineable data from medical images using advanced feature analysis [92,93]. Radiomics allows medical personnel to extract and analyze quantitative features from medical images to predict tumor behavior, treatment response, and patient outcomes [94]. This leading-edge paradigm has exhibited competencies in determining and grading glioma tumors. It has also been utilized in survival prediction [95] and has paved the way for precision medicine in these tumors [96]. Traditionally, radiomics features include shape, histogram (intensity/density), and texture features, often called handcraft features. More recently, the features extracted by deep learning models have a robust research momentum, often called deep features. Considering radiomics as an invaluable tool in the diagnosis, treatment, and prognosis of brain tumors, radiomics has proven to differentiate high-grade gliomas versus tumefactive demyelinating diseases significantly [97,98]. Notably, radiomics has shown practical applications in pretreatment evaluation, prognosis, survival, and post-treatment evaluation of glioma and GBM. Regarding the pretreatment assessment, radiomics may detect infiltration and the extent of brain tumors [99,100]. However, its use in differentiating radionecrosis and pseudoprogression from the true progression and recurrence of these tumors is a controversial debate. The present review aims to explore the current radiomics approaches for determining these conditions. Moreover, it delineates a body of knowledge supporting radiomics’s effectiveness in differentiating these scenarios.

## 3. Radiomics Differentiates

The detailed information on our reviewed studies is mentioned in Table 2.

### 3.1. Pseudoprogression vs. True Progression

When considering the use of radiomics in the post-treatment assessment of glioma, pseudoprogression (PSP) and distinguishing residual/recurrent tumors from post-treatment changes are among the primary concerns of researchers. In one study, 61 patients with glioblastoma were studied using conventional MRI, DWI, and PWI to distinguish between PSP and TP within three months of radiochemotherapy and surgical resection. Radiomics features were extracted from postcontrast T1, FLAIR, and ADC, and the LASSO model selected CBV maps as the most predictive feature. Subsequently, 12 features were used for the machine learning model, including two postcontrast T1, one FLAIR, two ADC, and seven CBV features. The final model had the area under the curve (AUC) of 0.9 to differentiate the PSP from TP. The AUC in the external validation dataset is 0.85. This multiparametric MRI radiomics was superior to ADC and CBV parameters alone [101].

However, in another study about GBM in 51 cases of TP and 26 cases of PSP, radiomics texture analysis of enhancing lesions (only T1 + C was used) scored much lower values, with accuracy (ACC), sensitivity, and specificity of 72.78%, 78.36%, and 61.33% in the differentiation of pseudoprogression from actual progression GBM [102]. Interestingly, this random forest (RF)–radiomics model still excelled in the performance of three radiologists. No clinical information, including sex, age, KPS score, resection extent, neurological deficit, and mean radiation dose, showed statistically significant differences between true progression and pseudoprogression.

**Table 2 cancers-15-04429-t002:** Tumoral and imaging characteristics of included studies.

Author (Year) (Country)	Sample Size (Mean Age)	Tumor Characteristics	Grade	Intervention	Follow-Up	Imaging Modalities	Tracer or Contrast (Dosage)
Zhang et al., (2019) (China) [103]	51 (47.6)	Glioblastoma 12, astrocytoma 14, ependymoma 3, mixed glioma 22	High (III–IV) 32 (62.7%), low (I–II) 19 (37.3%)	Radiation and surgery	>6 months	3.0 T MRI T1/T1C/T2/FLAIR	Gadolinium (0.1 mmol/kg)
Tiwari et al., (2016) (USA) [104]	43 (N/M)	Glioma 33, metastasis 25	-	At least one patient had surgical resection	>9 months	T1 + C WI/T2WI/FLAIR	Gadolinium (-)
Gao et al., (2020) (China) [105]	39 (51.45)	Glioblastomas and anaplastic astrocytoma	Glioma recurrence: (III = 4, IV = 21) TRE: (III = 3, IV = 11)	Radiotherapy and chemotherapy	>6 months	3.0 T MRI T1WI/T2 FLAIR, postcontrast T2FLAIR	Gadopentetate dimeglumine (0.1 mmol/Kg)
Chen et al., (2015) (China) [106]	22 (43.54)	Glioblastoma	-	Surgical resection + CCRT with TMZ	6 months	1.5 T MRI T1WI, T2WI, FLAIR, and T1Ce	Gadobutrol (0.1 mmol/kg)
Sadique et al., (2022) (USA) [107]	30 (N/M)	Glioblastoma (from different public datasets)	-	-	2–3 months	T1, T2, FLAIR, T1 + C	-
Wang et al., (2019) (China) [108]	160 (44.59)	Glioma	-	Radiation therapy + TMZ + six cycles of adjuvant TMZ	40 months	18F-FDG, 11C-MET, and 3.0 T MRI (T1 + C and FALIR)	18F-FDG (3.7 MBq/kg) 11C-MET PET (555–740 MBq)
Sun et al., (2021) (China) [102]	77 (49.1)	Glioblastoma	-	Total resection or subtotal resection + CCRT and TMZ	6 months	3.0 T MRI (T1 + C)	Gadodiamide (0.1 mmol/kg)
Hotta et al., (2019) (Japan) [109]	41 (55.5)	Glioma	Grade 2 (n = 4), grade 3 (*n* = 8), and grade 4 (*n* = 8)	Radiation therapy (either conventional radiotherapy or stereotactic radiosurgery)	6 months	PET/CT	MET (384.0 ± 22.7 MBq)
Park et al., (2021) (S. Korea) [110]	127 (57.46)	Grade 4 GBM + R132H mutation in IDH1, MGMT	-	Surgery with chemoradiation	2–3 months	3.0 T MRI T1, T2, ADC	Gadolinium (0.1 mL/kg)
Jiang et al., (2022) (USA) [111]	86 (52.15)	Primary malignant glioma (glioblastoma-anaplastic, oligodendroglioma-anaplastic, astrocytoma-gliosarcoma)	Recurrence: (III = 22, IV = 38), treatment: (III = 4, IV = 22)	Gross total resection, other surgical procedures + chemoradiation or radiotherapy	Range, 18 days to 3655 days	3.0 T MRI (APTw)—T2w, FLAIR, T1w, and (Gd-T1w)	Gadolinium (0.2 mL/kg)
Zhang et al., (2022) (China) [112]	126 (46.25)	Grades 2–4 GBM	-	Surgery + chemoradiation or radiotherapy	2–3 months	3.0 T MRI T_1_WI, T_2_WI, T_2_FLAIR, DWI, ASL, and CE-T_1_WI	-

Abbreviations: T1C: postcontrast T1-weighted sequence; FLAIR: fluid-attenuated inversion recovery; CCRT with TMZ: concurrent chemoradiotherapy with temozolomide; FDG: fluorodeoxyglucose; MET: methionine; MRI: magnetic resonance imaging; PET: positron emission tomography; GBM: glioblastoma multiform; IDH1: isocitrate dehydrogenase 1; ADC: apparent diffusion coefficient; APTw: amide proton transfer-weighted.

### 3.2. Recurrent Brain Tumor vs. Radiation Necrosis

In one feasibility study on 33 glioma patients, radiomics features from T1 + C, T2, and FLAIR were compared by neuroradiologists for differentiation between RN and tumor recurrence (TR). These features had a superior performance with an AUC of 0.79, while FLAIR was the most crucial sequence [104].

In one study on 16 patients with RN and 35 with TR, the researchers extracted handcraft and deep features and used them to differentiate RN versus TR. They used T1, T1 + C, T2, and FLAIR sequences. They showed that handcraft multiparametric MRI features were superior to single-sequence radiomics with an AUC of 0.96. Also, they showed that adding deep features to handcraft features can improve the model’s performance up to an AUC of 0.99. Of note, three patients in this study had ependymoma, and the rest had glioma [103].

In another study (training data set: 63 recurrent GBM and 23 RN; external validation: 23 recurrent GBM and 18 RN), T1 + C and T2 were compared to the ADC. The study concluded that ADC radiomics is superior to T1 + C and T2 and announced ADC to be more capable of differentiating recurrent GBM from TN with an AUC, accuracy, sensitivity, and specificity of 0.80, 78%, 66.7%, and 87%, respectively [110]. Diffusion radiomics models seem promising in these scenarios since they reflect tumor microenvironments [110]. Moreover, second-order features were the most prominent features extracted from the diffusion model [110]. However, quantitative features such as flatness, sphericity, mesh volume, and significant axis length aided this study [110].

In one study about 11C-methionine (MET)-PET on brain metastasis (15 TR, 6 RN) and 23 gliomas (18 TR, 5 RN), radiomics analysis had an AUC of 0.98, which was significantly higher than the traditional evaluation of lesions on brain PET (ratio of tracer uptake in tumor-to-normal-cortex; T/N ratio) [109].

In one radiomics study on 118 recurrent gliomas and 42 RT, T1 + C, and FLAIR MRI, 18F-fluorodeoxyglucose (18F-FDG) positron emission tomography (PET), and 11C-methionine (11C-MET) PET were compared. For a single modality, the AUC values of 0.81, 0.75, and 0.62 were reported for 18F-FDG, 11C-MET, and MRI, respectively, while the combination of 18F-FDG + 11C-MET, 18F-FDG + MRI, and 11C-MET + MRI had AUC values of 0.89, 0.86, and 0.8, respectively [108].

In a study about 11C-methionine (MET)-PET, the tumor-to-normal-cortex (T/N) ratio >1.3 was used as the traditional method for differentiation between TR and RN. This threshold was then compared with radiomics analysis (42 features from the lesion with elevated uptake). For this task, radiomics significantly outperformed the T/N ratio (sensitivities of 90.1% and 60.6%, specificities of 93.9% and 72.7%, and AUC of 0.98 and 0.73, respectively). GLCM has also been considered the most discriminative feature when using PET-MET in cases of recurrent brain tumor vs. radiation necrosis (RN). This may be due to higher intratumoral heterogeneity of the brain tumor and indicates the importance of heterogeneity over intensity uptake [109]. Histopathologically, pseudoprogression exhibits necrotizing effects with a complete absence of tumor cells, vascular dilation, fibrinoid necrosis, and normal cerebral vasculature endothelial damage. At the same time, true progression presents with tumor cells, increased cellularity, and vascular proliferation, except for necrotizing treatment effects. This indicates a more heterogenous pathologic nature of tumor progression and pinpoints the importance of methods identifying the differences in underlying tissue heterogeneity [106].

An amide proton transfer-weighted (APTw-MRI) radiomics model exhibited 86.0% accuracy in diagnosing tumor recurrence from radiotherapy effects, outperforming radiomics analysis on single sequences of T1, T1 + C, T2, and FALIR. The accuracy of this model rose to 89.5% when combined with structural MR images (T1 w, T 2 w, FLAIR, Gd-T 1 w) [111]. The APTw signal intensity originates from amide protons in endogenous proteins and peptides in the parenchyma. In tumor tissue, the content of mobile proteins and peptides is higher than in normal tissue, resulting in increased APTw signal intensity [111,113].

One recent study (66 TR and 30 RN for the training dataset and 18 TR and 12 RN for the validation dataset) evaluated the performance of traditional MR sequences (T1, T2, FLAIR, T1 + C), DWI, and ASL (CBF) for differentiation between TR and RN in glioma. An SVM model which demonstrated a multiparameter MRI radiomics model (AUC 0.96) was superior to conventional MRI (AUC 0.88), ADC (AUC 0.91), and ASL-CBF (AUC 0.95) radiomics models. Also, the multiparameter MRI radiomics model had an AUC of 0.98 in WHO grade 2~3 and 0.96 in WHO grade 4 [112]. In this study, incorporating ASL modality contributed to differentiating recurrence from radiation-induced lesions more than other modalities [112]. In addition, this study used WHO 2021 guidelines as the inclusion criteria and is probably the most reliable publication available now.

The reviewed studies aimed to develop and determine machine learning models’ capacity in differentiating tumor necrosis and progression from treatment (radiotherapy)-related complications. The details of the studies’ characteristics, imaging modalities, and radiomics procedures are mentioned in Table 1, Table 2 and Table 3 at the end of the manuscript.

## 4. Limitations and Challenges

In 2021, the World Health Organization (WHO) introduced a new classification system for glioma and glioblastoma. The new system categorizes diffuse gliomas into four general groups: adult-type diffuse gliomas, pediatric-type diffuse low-grade gliomas, pediatric diffuse high-grade gliomas, and circumscribed astrocytic gliomas. The updated classification system more strictly defines GBM as a highly malignant tumor. Additionally, the new design includes updated grading within tumor types [114]. Most published studies about radiomics in post-treatment glioma used older 2016 classification, and their results are not necessarily generalized to current glioma classification.

Although some studies used data from pathologically confirmed patients [108,112], most of them used clinical and radiologic findings as the mainstay of diagnosis instead [105]. In this context, the enhancing lesions within the resection cavity and at the radiation field are considered TR if they show enlargement on subsequent imaging and post-treatment if they get smaller on follow-up.

A small data set was a common downfall among studies [103,104,106], which can result in positive bias, lower statistical power, and unreliable results and can damage the generalizability of the findings [115,116]. To overcome this obstacle, some studies used five-fold and 10-fold cross-validation techniques to evaluate their machine learning models [107]. Some others used an independent validation cohort, which provides an unbiased estimate of the model’s performance on unseen data [104]. However, most published studies used just internal datasets, and their results are subject to overestimation of the performance of radiomics.

Many of these studies were conducted retrospectively [105], which may have been damaged by selection bias, recall bias, less validity, and more common errors [117].

The enhancing region within the resection cavity and in the radiation field is the most common region analyzed by radiomics, limiting the assessment of nonenhanced tumor infiltration areas. Thus, multimodal parameters reflecting nonenhanced lesions should be considered in future studies [105].

In some studies, radiologists manually contoured lesions [106]. This may lead to observer bias, high inter-reader variability, the derivation of unstable radiomics features, and increased variability in image acquisition and reconstruction, which can affect the reproducibility of radiomics features [93,118,119]. ITK-SNAP (http://www.itksnap.org/) was the main segmentation software used in studies, although other software was employed, too. Notably, using different segmentation software can introduce variability and affect the reproducibility and accuracy of radiomics features [93] (Table 2).

Lesions with a small volume of interest or trace uptake were excluded from the analysis of some studies. Although it may be inaccessible to fully consider all the regions, with improvement in processing and analysis methods, these regions can be explored and increase the accuracy of studies [109].

A large and varied dataset for training is one of the most critical challenges in model creation for AI. To address this, developing public datasets as benchmarks for AI algorithms is important. It is essential to consider that radiomics can be affected by differences in MRI machines, magnet strength, and acquisition parameters. While models are trained using data from multiple scanners for greater accuracy, there is uncertainty about how well they will work on new datasets acquired from different scanners with diverse scanning protocols. Preprocessing methods such as brain extraction, denoising, segmentation, and feature extraction are essential in medical imaging, but standardization proves challenging due to different criteria and parametrization. To ensure consistency, it is essential to document acquisition protocols and preprocessing pipelines in detail [120].

Despite the benefits of sharing patient data, there are still several challenges. First, it can be difficult to distribute these data, particularly multicenter datasets, when the patient data take up a lot of storage space, for example, with high-resolution images. Second, there are legal and ethical barriers that prevent the public distribution of some or all of the data due to privacy concerns [121]. Third, institutions may choose not to share patient data, considering the data an invaluable asset [122]. Currently, most of the radiomics-trained models about post-treatment glioma have yet to be publicly available. That means that other researchers cannot adopt them for further improvement.

## 5. Conclusions

The preliminary data are auspicious in applying the radiomics model in post-treatment glioma follow-up imaging and to differentiate between pseudoprogression from true progression and radiation necrosis from tumor recurrence, often with reported AUC above 0.9. Radiomics can be used on traditional MR sequences (T1, T2, FALIR, T1 + C, DWI/ADC), advanced MR sequences (MR perfusion, APTw-MRI), as well as PET (18F-FDG and 11C-MET). The performance of radiomics is likely superior to traditional radiologist diagnosis. Also, the performance of radiomics is enhanced by using multiparametric images (adding different MR sequences, ADC, MR perfusion, and PET) compared to a single modality. However, limited data sets, different preprocessing and analysis software, and methods still need to be improved in reaching a consensus on the exact paradigms of this novel method (Table 3). Further prospective investigations with larger data sets and varied imaging modalities are in huge demand.

**Table 3 cancers-15-04429-t003:** Radiomics features of included studies.

	Machine Learning Features			Radiomics Features		Performance Metrics						
Author (Year)	Model	Method of Measuring Performance	Segmentation Software # Feature Extraction and Selection Software # Feature Analysis Software		Type of Features # Number of Selected and Extracted Features	Modality	Sensitivity	Specificity	PPV	NPV	AUC	Accuracy
Zhang et al., (2019) (China) [103]	Random forest, naive Bayes classifiers, AlexNet, Inception v3 CNN	Cross-validation	ITKSNAP software, FSL5.0.9 # AlexNet, and Inception v3 # MATLAB 2017b		Handcrafted, deep texture features # 4 nontexture features, 41,284 texture features, 16,384 AlexNet features, and 8192 Inception v3 features extracted	Multimodality MRI	99.4%	97.5%	NA	NA	0.998	97.8%
Tiwari et al., (2016) (USA) [104]	SVM classifier	Independent validation cohort	3D Slicer and BraTumIA # Matlab R 2014b # Matlab R2014b		Spatial distribution of pixel intensities within the MRI images and included features # E: 119 2D texture features on a per-voxel basis # S: 78	FLAIR	NA	NA	NA	NA	0.79	75%
						T2	NA	NA	NA	NA	0.77	72%
Gao et al., (2020) (China) [105]	SVM classifier	Five-fold cross-validation	ITK-SNAP # E: PyRadiomics S: recursive feature elimination (RFE) # N/A		Three first-order features, eight gray-level co-occurrence matrix (GLCM) features, and two gray-level run-length (GLRLM) features # E: 186 S: 13	T1 + C	100%	70%	62.5%	100%	0.8	80%
						T2 FLAIR + C	100%	80%	71.43%	100%	0.84	86.67%
						T1C subtraction + T2 FLAIR subtraction	100%	90%	83.33%	100%	0.94	93.33%
Chen et al., (2015) (China) [106]	N/A	N/A	Manually # N/A # MedCalc IBM SPSS Statistics		GLCM texture # N/A	T1 + C	91.7%	70%	78.6%	87.5%	0.84	81.8%
						T2	75%	100%	66.7%	100%	0.88	86.4%
						FLAIR	66.7%	80%	80%	66.7%	0.75	72.7%
Sadique et al., (2022) (USA) [107]	Random forest (RF) classifier	Stratified five-fold cross-validation, leave-one-out cross-validation	A 3D deep learning model was used to segment subregions of the tumor, which were verified by a radiologist # N/A # N/A		Multiresolution texture features, texture features #	Texture, volumetric, and histogram features	NA	94%	NA	NA	NA	93%
Wang et al., (2019) (China) [108]	Computer-supported predictive models	Cross-validation	ITK-SNAP # AnalysisKit (GE Healthcare, China) # R studio		First-order features, shape features, and texture features # E: 912 (FDG 303; MET 297; MRI 312) # S: 8–13	FDG	69%	76%	NA	NA	0.8	71%
						MET	75%	69%	NA	NA	0.75	73%
						MRI	62%	65%	NA	NA	0.62	69%
						FDG + MET	75%	91%	NA	NA	0.89	79%
						FDG + MRI	83%	75%	NA	NA	0.86	81%
						MET + MRI	72%	58%	NA	NA	0.8	68%
Sun et al., (2021) (China) [102]	Random forest classifier	RF classifier trained with 50 trees, 10-fold cross-validation	ITK-SNAP version 3.6 # E: Analysis-Kinetics (A.K., GE Healthcare) S: R version 3. 4. 2 # SPSS 20		42 histogram features, 11 Gy-level size zone matrix (GLSZM) texture features, 10 Haralick features, 144 Gy-level co-occurrence matrix (GLCM) texture features, and 180 run-length matrix (RLM) texture features # E: 9675 S: 50	T1 + C	78%	61%	NA	NA	NA	72%
Hotta et al., (2019) (Japan) [109]	Random forest classifier	10-fold cross-validation	LIFEX # E: LIFEx S: R package “Boruta” # LIFEx		Texture features extracted from MET-PET images using gray-level co-occurrence matrix (GLCM), gray-level run-length matrix (GLRLM), gray-level zone-length matrix (GLZLM), and neighborhood gray-level difference matrix (NGLDM) # E: 42 S: 30	Radiomics	90.1%	93.9%	95.2%	88.6%	0.98	92.2%
						T/N Ratio	60.6%	72.7%	86.9%	38.1%	0.73	63.6%
Park et al., (2021) (S. Korea) [110]	SVM, KNN, AdaBoost	10-fold cross-validation	3D slicer (semiautomatic) # PyRadiomics # R-WhiteStripe		GLCM, GLRLM, GLSZM, NGTDM # E: 263 S: 18	LASSO feature selection and SVM	66.7%	87%	NA	NA	0.8	78%
Jiang et al., (2022) (USA) [111]	N/A	N/A	ITK-SNAP # PyRadiomics # SPSS 26.0 MATLAB R2021a		APTw features # E: 525 for each sequence; a total of 2589 S: for T1w, T2w, FLAIR, Gd-T1w, or APTw MR images, 34, 61, 47, 18, or 176 radiomics features were selected	All sequences	85%	100%	NA	NA	92.5%	89.5%
						APTw	70.6%	96.2%	NA	NA	87.8%	86%
						T1	96.7%	23.1%	NA	NA	59.9%	74.4%
						T2	58.3%	90%	NA	NA	77.9%	76.7%
						FLAIR	88.3%	73.1%	NA	NA	80.7%	83.7%
						T1 + C1	0	75%	NA	NA	61.5%	76.7%
						T1, T2, FLAIR, T1 + C8	5%	76.9%	NA	NA	81%	82.6%
						T1, T2, FLAIR, APTw	88.3%	96.2%	NA	NA	92.2%	90.7%
Zhang et al., (2022) (China) [112]	SVM, KNN, LR, NB	10-fold cross-validation	ITK-SNAP (Manually) # MATLAB # MATLAB, Python 3.8		GLCM, GLRLM, GLSZM, first order # E: 4199 S: eight (two T1, one T1 + C, one ADC, four CBF)	SVM and multiparameter MRI	91.7%	NA	NA	NA	0.94	NA
							100%	NA	NA	NA	0.82	NA

Abbreviations: PPV: positive predictive value; NPV: negative predictive value; AUC: area under the curve; CNN: convolutional neural network; SVM: support vector machine; GLCM: gray-level co-occurrence matrix; GLRLM: gray-level run-length matrix; RF: random forest; GLSZM: gray-level size zone matrix; RLM: run-length matrix; NGLDM: neighborhood gray-level difference matrix; KNN: k-nearest neighbors algorithm; APTw: amide proton transfer-weighted; LR: logistic regression; NB: naive Bayes; CBF: cerebral blood flow.

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
