# Peer review of "Radiomics: The New Promise for Differentiating Progression, Recurrence, Pseudoprogression, and Radionecrosis in Glioma and Glioblastoma Multiforme"

_cancers, 2023, doi:10.3390/cancers15184429_

Round 1

Reviewer 1 Report

Authors performed a good review however it is not effectively transferred to the text based on the table design. So, in my opinion, most of the text of tables should be included in the manuscript for the table 1.

Table 2 should be completely modify and size reduced in order to make it more easy to understand and dynamic. For example, in sample size are included results of RP y RNP that have not so much sense, WHO 2021 only is used on papers after this data, so it can be eliminate and the same with the timing and brand. Try to unify and reduce to the minimum number. Think that a table in landscape orientation of more than 2 pages is too much big and with difficulties in the visual identification of data. After reduce the number of columns, is someone keep not filled, think to eliminate it.

Table 2 of page 13 is the table 3. The same explanations that in the previous table...a lot of subdivisions. In statistical analysis is included feature analysis software that corresponds to radiomic features section. So radiomic features should be an unique column and the same for the rest of sections as the results or performance metrics.

All the references  should be deeply revised because there are a lot of duplications (116-125, 111-128,113-124, among others...).

English grammar should be revised in all the text and tables a corrected the several edition mistakes. 

Acronyms should be revised and introduced in all the text and thereafter used (for example in pseudo progression)

Preferably do not use acronym in paragraph titles . 

Author Response

Point 1: Authors performed a good review however it is not effectively transferred to the text based on the table design. So, in my opinion, most of the text of tables should be included in the manuscript for the table 1.

Response 1: First of all, we all sincerely thank you for your thoughtful comments, which have contributed to the improvement of our manuscript. Table 1 is wholly transformed into a new table, only containing performance metrics of each imaging modality in diagnosis post-treatment glioma. We have transferred their application and limitation debates into the manuscript text.

Point 2: Table 2 should be completely modified and size reduced in order to make it more easy to understand and dynamic. For example, in sample size are included results of RP y RNP that have not so much sense, WHO 2021 only is used on papers after this data, so it can be eliminate and the same with the timing and brand. Try to unify and reduce to the minimum number. Think that a table in landscape orientation of more than 2 pages is too much big and with difficulties in the visual identification of data. After reduce the number of columns, is someone keep not filled, think to eliminate it.

Response 2: Thank you for your comment. Table 2 has been modified and reduced in size. We have merged all the participants’ numbers as a whole count. Also, unnecessary and incomplete data regarding the WHO 2021 classification, scanning timing, and brand of the scanner have been removed.

Point 3: Table 2 of page 13 is the table 3. The same explanations that in the previous table...a lot of subdivisions. In statistical analysis is included feature analysis software that corresponds to radiomic features section. So radiomic features should be an unique column and the same for the rest of sections as the results or performance metrics.

Response 3: Thank you for your comment. Table 3 name has been corrected and modified in size and visualization. The sections for the “region of interest”, “Normalization and scaling methods”, “regression model”, and “feature selection criteria” have been removed since they had little relevance to the concept of the manuscript. Therefore, their elimination could bring more clear, concise and visually appealing findings at the expense of losing less critical data. Further, feature analysis software, wrongly categorized under the statistical analysis section, has been taken under radiomic features classification. Thus, the radiomic features section now has two main subdivisions: a box dedicated to all the software used in the radiomics process and another box describing types and numbers of extracted and selected features.

Point 4: All the references should be deeply revised because there are a lot of duplications (116-125, 111-128,113-124, among others...).

Response 4: Thank you for your comment. References have been deeply revised, and duplicates have been eliminated.

Comments on the Quality of English Language

Point 5: English grammar should be revised in all the text and tables a corrected the several edition mistakes. 

Response 5: Thank you for your comment. English grammar has been reviewed, and the overall quality of the text has been enhanced.

Point 6: Acronyms should be revised and introduced in all the text and thereafter used (for example in pseudo progression)

Response 6: Thank you for your comment. The complete form of each abbreviation has been mentioned as the first time it appeared.

Point 7: Preferably do not use acronym in paragraph titles. 

Response 7: Thank you for your comment. Acronyms used in the paragraph titles have been substituted with an expanded version of the word.

All the changes made during this revision are marked with track change feature in Microsoft Office Word.

Reviewer 2 Report

The authors present a paper about "Radiomics: The New Promise for Differentiating Progression, Recurrence, Pseudo-Progression, and Radio-necrosis in Glioma and GBM"

The literature search is by far comprehensive and the topic totally deserves attentions from the clinical and scientific point of view.

I have some concers that I would like the authors to address as follows:

1) In the title the author talk about "Radiomics: The New Promise for Differentiating Progression, Recurrence, Pseudo-Progression, and Radio-necrosis in Glioma and GBM": how do they define the difference between Progression and Recurrence? is there a time cut-off? In the text they talk toghether about "Recurrence/True Progression". They should clarify if they just want to talk about progression or recurrence because froma clinical point of view they are two different topics

2) In the text they talk about "Pseudo-response": is this point relevant? if the authors believe it is relevant they should focus ore in detail in the text otherwise I my opinion it might also possible not to mention it in the article

3) It would be interesting to have some additional details with regard to the technical features of the MRI machines used (1.5 Tesla? 3 Tesla? 7 Tesla?)  

Author Response

The authors present a paper about "Radiomics: The New Promise for Differentiating Progression, Recurrence, Pseudo-Progression, and Radio-necrosis in Glioma and GBM". The literature search is by far comprehensive and the topic totally deserves attentions from the clinical and scientific point of view. I have some concerns that I would like the authors to address as follows:

Point 1: In the title the authors talk about "Radiomics: The New Promise for Differentiating Progression, Recurrence, Pseudo-Progression, and Radio-necrosis in Glioma and GBM": how do they define the difference between Progression and Recurrence? is there a time cut-off? In the text they talk together about "Recurrence/True Progression". They should clarify if they just want to talk about progression or recurrence because from a clinical point of view, they are two different topics.

Response 1: First of all, we all sincerely thank you for your thoughtful comments, which have contributed to the improvement of our manuscript. Regarding the recurrence/true progression, we first mentioned them beside each other to showcase their similar manifestation that confounds the diagnosis of radiation necrosis. However, further in the text, we described them as separate entities based on your recommendation.

Point 2: In the text they talk about "Pseudo-response": is this point relevant? if the authors believe it is relevant, they should focus ore in detail in the text otherwise I my opinion it might also possible not to mention it in the article.

Response 2: Thank you for your comment. The “Pseudo-response” section has been removed since it shared little relevant value with the rest of the manuscript.

Point 3: It would be interesting to have some additional details with regard to the technical features of the MRI machines used (1.5 Tesla? 3 Tesla? 7 Tesla?).

Response 3: Thank you for your comment. The Tesla power of MRI modalities was mentioned in Table 2, whenever available.

All the changes made during this revision are marked with track change feature in Microsoft Office Word.

Round 2

Reviewer 1 Report

Thanks a lot for all the work performed.

Thanks a lot for all the changes introduced 

Reviewer 2 Report

All my previous comments have been satisfactorily addressed by the authors